# k-Prototype Learning for 3D Rigid Structures $^{\star}$

**Hu Ding**
Department of Computer Science and Engineering
State University of New York at Buffalo
Buffalo, NY14260
huding@buffalo.edu

**Ronald Berezney**
Department of Biological Sciences
State University of New York at Buffalo
Buffalo, NY14260
berezney@buffalo.edu

**Jinhui Xu**
Department of Computer Science and Engineering
State University of New York at Buffalo
Buffalo, NY14260
jinhui@buffalo.edu

## Abstract

In this paper, we study the following new variant of prototype learning, called *k-prototype learning problem for 3D rigid structures*: Given a set of 3D rigid structures, find a set of $k$ rigid structures so that each of them is a prototype for a cluster of the given rigid structures and the total cost (or dissimilarity) is minimized. Prototype learning is a core problem in machine learning and has a wide range of applications in many areas. Existing results on this problem have mainly focused on the graph domain. In this paper, we present the first algorithm for learning multiple prototypes from 3D rigid structures. Our result is based on a number of new insights to rigid structures alignment, clustering, and prototype reconstruction, and is practically efficient with quality guarantee. We validate our approach using two type of data sets, random data and biological data of chromosome territories. Experiments suggest that our approach can effectively learn prototypes in both types of data.

## 1 Introduction

Learning prototype from a set of given or observed objects is a core problem in machine learning, and has numerous applications in computer vision, pattern recognition, data mining, bioinformatics, etc. A commonly used approach for this problem is to formulate it as an optimization problem and determine an object (called *pattern* or *prototype*) so as to maximize the total similarity (or minimize the total difference) with the input objects. Such computed prototypes are often used to classify or index large-size structural data so that queries can be efficiently answered by only considering those prototypes. Other important applications of prototype include reconstructing object from partially observed snapshots and identifying common (or hidden) pattern from a set of data items.

In this paper, we study a new prototype learning problem called *k-prototype learning* for 3D rigid structures, where a 3D rigid structure is a set of points in $R^3$ whose pairwise distances remain invariant under rigid transformation. Since our problem needs to determine $k$ prototypes, it thus can be viewed as two tightly coupled problems, clustering rigid structures and prototype reconstruction for each cluster.

Our problem is motivated by an important application in biology for determining the spatial organization pattern of chromosome territories from a population of cells. Recent research in biology [3]

---

$^{\star}$This research was supported in part by NSF under grant IIS-1115220.

has suggested that configuration of chromosome territories could significantly influence the cell molecular processes, and are closely related to cancer-promoting chromosome translocations. Thus, finding the spatial organization pattern of chromosome territories is a key step to understanding the cell molecular processes [6,7,10,25]. Since the set of observed chromosome territories in each cell can be represented as a 3D rigid structure, the problem can thus be formulated as a $k$-prototype learning problem for a set of 3D rigid structures.

**Related work:** Prototype learning has a long and rich history. Most of the research has focused on finding prototype in the graph domain. Jiang *et al.* [18] introduced the *median graph* concept, which can be viewed as the prototype of a set of input graphs, and presented a genetic approach to solve it. Later, Ferrer *et al.* [14] proposed another efficient method for median graph. Their idea is to first embed the graphs into some metric space, and obtain the median using a recursive procedure. In the geometric domain, quite a number of results have concentrated on finding prototypes from a set of 2D shapes [11,20,21,22]. A commonly used strategy in these methods is to first represent each shape as a graph abstraction and then compute the median of the graph abstractions.

Our prototype learning problem is clearly related to the challenging 3D rigid structure clustering and alignment problem [1,2,4,5,13,17]. Due to its complex nature, most of the existing approaches are heuristic algorithms and thus cannot guarantee the quality of solution. There are also some theoretical results [13] on this problem, but none of them is practical due to their high complexities.

**Our contributions and main ideas:** [1] Our main objective on this problem is to obtain a practical solution which has guarantee on the quality of its solution. For this purpose, we first give a formal definition of the problem and then consider two cases of the problem, 1-prototype learning and $k$-prototype learning.

For 1-prototype learning, we first present a practical algorithm for the alignment problem. Our result is based on a *multi-level net* technique which finds the proper Euler angles for the rigid transformation. With this alignment algorithm, we can then reduce the prototype learning problem to a new problem called *chromatic clustering* (see Figure 1(b) and 1(c) ), and present two approximate solutions for it. Finally, a local improvement algorithm is introduced to iteratively improve the quality of the obtained prototype.

For $k$-prototype learning, a key challenge is how to avoid the high complexity associated with clustering 3D rigid structures. Our idea is to map each rigid structure to a point in some metric space and build a *correlation graph* to capture their pairwise similarity. We show that the correlation graph is metric; this means that we can reduce the rigid structure clustering problem to a metric $k$-median clustering problem on the correlation graph. Once obtaining the clustering, we can then use the 1-prototype learning algorithm on each cluster to generate the desired prototype. We also provide techniques to deal with several practical issues, such as the unequal sizes of rigid structures and the weaker metric property caused by imperfect alignment computation for the correlation graph.

We validate our algorithms by using two types of datasets. The first is randomly generated datasets and the second is a real biological dataset of chromosome territories. Experiments suggest that our approach can effectively reduce the cost in prototype learning.

## 2  Preliminaries

In this section, we introduce several definitions which will be used throughout this paper.

**Definition 1** (*m*-**Rigid Structure**). *Let $P = \{p_1, \cdots, p_m\}$ be a set of $m$ points in 3D space. $P$ is an $m$-rigid structure if the distance between any pair of vertices $p_i$ and $p_j$ in $P$ remains the same under any rigid transformation, including translation, rotation, reflection and their combinations, on $P$. For any rigid transformation $\mathcal{T}$, the image of $P$ under $\mathcal{T}$ is denoted as $\mathcal{T}(P)$.*

**Definition 2** (**Bipartite Matching**). *Let $S_1$ and $S_2$ be two point-sets in 3D space with $|S_1| = |S_2|$, and $G = (U, V, E)$ be their induced complete bipartite graph, where each vertex in $U$ (or $V$) corresponds to a unique point in $S_1$ (or $S_2$), and each edge in $E$ is associated with a weight equal to the Euclidean distance of the corresponding two points. The bipartite matching of $S_1$ and $S_2$, is the one-to-one match from $S_1$ to $S_2$ with the minimum total matching weight (denoted as $Cost(S_1, S_2)$) in $G$ (see Figure 1(a)).*

Note that the bipartite matching can be computed using some existing algorithms, such as the *Hungarian algorithm* [24].

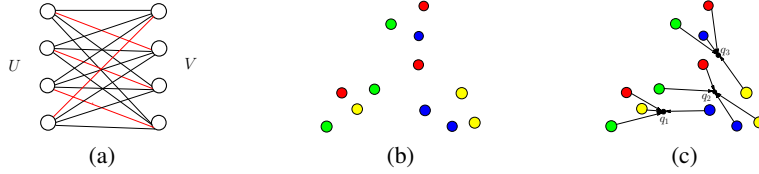

Fig. 1: (a) An example of bipartite matching (red edges); (b) 4 point-sets with each in a different color; (c) chromatic clustering of point-sets in (b). The three clusters form a chromatic partition.

**Definition 3 (Alignment).** *Let $P$ and $Q$ be two $m$-rigid structures in 3D space with points $\{p_1, \cdots, p_m\}$ and $\{q_1, \cdots, q_m\}$ respectively. Their alignment is to find a rigid transformation $\mathcal{T}$ for $P$ so as to minimize the cost of the bipartite matching between $\mathcal{T}(P)$ and $Q$. The minimum (alignment) cost, $\min_{\mathcal{T}} Cost(\mathcal{T}(P), Q)$, is denoted by $\mathcal{A}(P, Q)$.*

**Definition 4 ($k$-Prototype Learning).** *Let $P_1, \cdots P_n$ be $n$ different $m$-rigid structures in 3D, and $k$ be a positive integer. $k$-prototype learning is to determine $k$ $m$-rigid structures, $Q_1, \cdots, Q_k$, so as to minimize the following objective function*

$$\sum_{i=1}^{n} \min_{1 \le j \le k} \mathcal{A}(P_i, Q_j). \tag{1}$$

From Definition 4, we know that the $k$-prototype learning problem can be viewed as first clustering the rigid structures into $k$ clusters and then build a prototype for each cluster so as to minimize the total alignment cost.

## 3  1-**Prototype learning**

In this section, we consider the 1-prototype learning problem. We first overview the main steps of our algorithm and then present the details in each subsection. Our algorithm is an iterative procedure. In each iteration, it constructs a new prototype using the one from previous iteration, and reduces the objective value. A final prototype is obtained once the objective value becomes stable.

**Algorithm: 1-prototype learning**

1. Randomly select a rigid structure from the input $\{P_1, \cdots, P_n\}$ as the initial prototype $Q$.
2. Repeatedly perform the following steps until the objective value becomes stable.
   (a) For each $P_i$, find the rigid transformation (approximately) realizing $\mathcal{A}(P_i, Q)$.
   (b) Based on the new configuration (*i.e.,* after the corresponding rigid transformation) of each $P_i$, construct an updated prototype $Q$ which minimizes the objective value.

Since both of $2(a)$ and $2(b)$ reduce the cost, the objective value would always decrease. In the next two subsections, we discuss our ideas for Step $2(a)$ (alignment) and Step $2(b)$ (prototype reconstruction), respectively. **Note** that the above algorithm is different with *generalized procrustes analysis (GPA)* [15], since the points from each $P_i$ are not required to be pre-labeled in our algorithm, while for GPA every input point should have an individual index. This is also the main difficulty for this prototype learning problem.

### 3.1  Alignment

To determine the alignment of two rigid structures, one way is to use our recent theoretical algorithm for point-set matching [13]. For any pair of point-sets $P$ and $Q$ in $R^d$ space with $m$ points each, our algorithm outputs, in $O(\frac{1}{\epsilon^{d^2}} m^{2d+2} \log^{2d}(m))$ time, a rigid transformation $\mathcal{T}$ for $P$ so that the bipartite matching cost between $\mathcal{T}(P)$ and $Q$ is a $(1 + \epsilon)$-approximation of the optimal alignment cost between $P$ and $Q$, where $\epsilon > 0$ is a small constant. Applying this algorithm to our 3D rigid structures, the running time becomes $O(\frac{1}{\epsilon^9} m^8 \log^6(m))$. The algorithm is based on following key

idea. First, we show that there exist 3 "critical" points, called *base*, in each of $P$ and $Q$, which control the matching cost. Although the base cannot be explicitly identified, it is possible to obtain it implicitly by considering all 3-tuples of the points in $P$ and $Q$. The algorithm then builds an $\epsilon$-net around each base point to determine an approximate rigid transformation. Clearly, this theoretical algorithm is efficient only when $m$ is small. For large $m$, we use the following relaxation.

First, we change the edge weight in Definition 2 from Euclidean distance to squared Euclidean distance. The following lemma shows some nice property of such a change.

**Lemma 1.** *Let $P = \{p_1, \cdots, p_m\}$ and $Q = \{q_1, \cdots, q_m\}$ be two m-rigid structures in 3D space, and $\mathcal{T}$ be the rigid transformation realizing the minimum bipartite matching cost (where the edge weight is replaced by the squared Euclidean distance of the corresponding points in Definition 2). Then, the mean points of $\mathcal{T}(P)$ and $Q$ coincide with each other.*

Lemma 1 tells us that to align two rigid structures, we can first translate them to share one common mean point, and then consider only the rotation in 3D space. (Note that we can ignore reflection in the rigid transformation, as it can be captured by computing the alignment twice, one for the original rigid structure, and the other for its mirror image.) Using Euler angles and 3D rotation matrix, we can easily have the following fact.

**Fact 1** *Give any rotation matrix $A$ in 3D, there are 3 angles $\phi, \theta, \psi \in (-\pi, \pi]$, and three matrices, $A_1, A_2$ and $A_3$ such that $A = A_1 * A_2 * A_3$, where*

$$A_1 = \begin{bmatrix} 1 & 0 & 0 \\ 0 & \cos\phi & -\sin\phi \\ 0 & \sin\phi & \cos\phi \end{bmatrix} \quad , \quad A_2 = \begin{bmatrix} \cos\theta & 0 & \sin\theta \\ 0 & 1 & 0 \\ -\sin\theta & 0 & \cos\theta \end{bmatrix} \quad and \quad A_3 = \begin{bmatrix} \cos\psi & -\sin\psi & 0 \\ \sin\psi & \cos\psi & 0 \\ 0 & 0 & 1 \end{bmatrix}.$$

From the above Fact 1, we know that the main issue for aligning two rigid structures $P$ and $Q$ is to find three proper angles $\phi, \theta, \psi$ to minimize the cost. Clearly, this is a non-convex optimization problem. Thus, we cannot use existing convex optimization methods to obtain an efficient solution. One way to solve this problem is to build a dense enough $\epsilon$-net (or grid) in the domain $[-\pi, \pi]^3$ of $\phi, \theta, \psi$, and evaluate each grid point to find the best possible solution. Clearly, this will be rather inefficient when the number of grid points is huge. To obtain a practically efficient solution, our strategy is to generalize the idea of building a dense net to recursively building a sparse net, which is called ***multi-level net***. At each level, we partition the current searching domain into a set of smaller regions, which can be viewed as a sparse net, and evaluate some representative point in each of the smaller region to determine its likelihood of containing the optimal point. The recursion will only continue at the most likely $N$ smaller regions (for some well selected parameter $N \geq 1$ in practice). In this way, we can save a great deal of time for searching the optimal point in those unlikely regions. Below is the main steps of our approach.

1. Let $S$ be the current searching space, which is initialized as $[-\pi, \pi]^3$, and $t, N$ be two input parameters. Recursively perform the following steps until the best objective value in two consecutive recursive steps roughly remains the same.
    (a) Uniformly partition $S$ into $t$ disjoint sub-regions $S = S_1 \cup \cdots \cup S_t$.
    (b) Randomly select a representative point $s_i \in S_i$, and compute the alignment cost under the rotational matrix corresponding to $s_i$ via Hungarian algorithm.
    (c) Choose the top $N$ points with the minimum objective values from $\{s_1, \cdots, s_t\}$. Let $\{s_{t_1}, \cdots, s_{t_N}\}$ be the chosen points.
    (d) Update $S = \bigcup_{i=1}^N S_{t_i}$.
2. Output the rotation which yields the minimum objective value.

**Why not use other alignment algorithms?** There are several existing alignment algorithms for 3D rigid structures, and each suffers from its own limitations. For example, the *Iterative Closest Point* algorithm [4] is one of the most popular algorithms for alignment. However, it does not generate the one-to-one match between the rigid structures. Instead, every point in one rigid structure is matched to its nearest neighbor in the other rigid structure. This means that some point could match multiple points in the other rigid structure. Obviously, this type of matching cannot meet our requirement, especially in the biological application where chromosome territory is expected to match only one

chromosome. Similar problem also occurs in some other alignment algorithms [1,5,17]. Arun *et al.* [2] presented an algebraic approach to find the best alignment between two 3D point-sets. Although their solution is a one-to-one match, it requires that the correspondence between the two point-sets is known in advance, which is certainly not the case in our model. *Branch-and-bound (BB)* approach [16] needs to grow a searching tree in the parameter space, and for each node it requires estimating the upper and lower bounds of the objective value in the corresponding sub-region. But for our alignment problem, it is challenging to obtain such accurate estimations.

## 3.2 Prototype reconstruction

In this section, we discuss how to build a prototype from a set of 3D rigid structures. We first fix the position of each $P_i$, and then construct a new prototype $Q$ to minimize the objective function in Definition 4. Our main idea is to introduce a new type of clustering problem called *Chromatic Clustering* which was firstly introduced by Ding and Xu [12], and reduce our prototype reconstruction problem to it. We start with two definitions.

**Definition 5 (Chromatic Partition).** *Let $\mathcal{G} = \{G_1, \cdots, G_n\}$ be a set of $n$ point-sets with each $G_i$ consisting of $m$ points in the space. A chromatic partition of $\mathcal{G}$ is a partition of the $n \times m$ points into $m$ sets, $U_1, \cdots, U_m$, such that each $U_j$ contains exactly one point from each $G_i$.*

**Definition 6 (Chromatic Clustering).** *Let $\mathcal{G} = \{G_1, \cdots, G_n\}$ be a set of $n$ point-sets with each $G_i$ consisting of $m$ points in the space. The chromatic clustering of $\mathcal{G}$ is to find $m$ median points $\{q_1, \cdots, q_m\}$ in the space and a chromatic partition $U_1, \cdots, U_m$ of $\mathcal{G}$ such that $\sum_{j=1}^{m} \sum_{p \in U_j} ||p - q_j||$ is minimized, where $|| \cdot ||$ denotes the Euclidean distance.*

From Definition 6, we know that chromatic clustering is quite similar to $k$-median clustering in Euclidean space; the only difference is that it has the chromatic requirement, *i.e.,* the obtained $k$ clusters should be a chromatic partition (see Figure 1(b) and 1(c)).

**Reduction to chromatic clustering.** Since the position of each $P_i$ is fixed (note that with a slight abuse of notation, we still use $P_i$ to denote its image $\mathcal{T}(P_i)$ under the rigid transformation $\mathcal{T}$ obtained in Section 3.1), we can view each $P_i$ as a point-set $G_i$, and the new prototype $Q$ as the $k$ median points $\{q_1, \cdots, q_m\}$ in Definition 6. Further, if a point $p \in P_i$ is matched to $q_j$, then it is part of $U_j$. Since we compute the one-to-one match, $U_j$ contains exactly one point from each $P_i$, which implies that $\{U_1, \cdots, U_m\}$ is a chromatic partition on $\mathcal{G}$. Let $p_j^i$ be the one in $P_i \cap U_j$. Then the objective function in Definition 4 becomes

$$\sum_{i=1}^{n} \sum_{j=1}^{m} ||p_j^i - q_j|| = \sum_{j=1}^{m} \sum_{i=1}^{n} ||p_j^i - q_j|| = \sum_{j=1}^{m} \sum_{p \in U_j} ||p - q_j||, \tag{2}$$

which is exactly the objective function in Definition 6. Thus, we have the following theorem.

**Theorem 1.** *Step $2(b)$ in the algorithm of $1$-prototype learning is equivalent to solving a chromatic clustering problem.*

Next, we give two constant approximation algorithms for the chromatic clustering problem; one is randomized, and the other is deterministic.

**Theorem 2.** *Let $\mathcal{G} = \{G_1, \cdots, G_n\}$ be an instance of chromatic clustering with each $G_i$ consisting of $m$ points in the space.*

1. *If $G_l$ is randomly selected from $\mathcal{G}$ as the $m$ median points, then with probability at least $1/2$, $G_l$ yields a 3-approximation for chromatic clustering on $\mathcal{G}$.*

2. *If enumerating all point-sets in $\mathcal{G}$ as the $m$ median points, there exists one $G_{i_0}$, which yields a 2-approximation for chromatic clustering on $\mathcal{G}$.*

*Proof.* We consider the randomized algorithm first. Let $\{q_1, \cdots, q_m\}$ be the $m$ median points in the optimal solution, and $U_1, \cdots, U_m$ be the corresponding chromatic partition. Let $p_j^i = G_i \cap U_j$. Since the objective value is the sum of the total cost from all point-sets $\{G_1, \cdots, G_n\}$, by Markov inequality, the contribution from $G_l$ should be no more than 2 times the average cost with probability at least $1/2$, *i.e.,*

$$\sum_{j=1}^{m} ||p_j^l - q_j|| \leq 2\frac{1}{n} \sum_{i=1}^{n} \sum_{j=1}^{m} ||p_j^i - q_j||. \tag{3}$$

From (3) and triangle inequality, if replacing each $q_j$ by $p_j^l$, the objective value becomes

$$\sum_{i=1}^{n}\sum_{j=1}^{m}||p_j^i - p_j^l|| \le \sum_{i=1}^{n}\sum_{j=1}^{m}(||p_j^i - q_j|| + ||q_j - p_j^l||) \tag{4}$$

$$= \sum_{i=1}^{n}\sum_{j=1}^{m}||p_j^i - q_j|| + n \times \sum_{j=1}^{m}||q_j - p_j^l|| \le 3\sum_{i=1}^{n}\sum_{j=1}^{m}||p_j^i - q_j||, \tag{5}$$

where (4) follows from triangle inequality, and (5) follows from (3). Thus, the first part of the theorem is true. The analysis for the deterministic algorithm is similar. The only difference is that there must exist one point-set $G_{i_0}$ whose contribution to the total cost is no more than the average cost. Thus the constant in the right-hand side of (3) becomes 1 rather than 2, and consequently the final approximation ratio in (5) turns to 2. Note that the desired $G_{i_0}$ can be found by enumerating all point-sets, and selecting the one having the smallest objective value.  □

*Remark 1.* Comparing the two approximation algorithms, we can see a tradeoff between the approximation ratio and the running time. The randomized algorithm has a larger approximation ratio, but a linear dependence on $n$ in its running time. The deterministic algorithm has a smaller approximation ratio, but a quadratic dependence on $n$.

**Local improvement.** After finding a constant approximation, it is necessary to conduct some local improvement. An easy-to-implement method is the follows. Let $\tilde{Q} = \{\tilde{q}_1, \cdots, \tilde{q}_m\}$ be the initial constant approximation solution. Compute the bipartite matching between $\tilde{Q}$ and each $G_i$. This yields a chromatic partition $\{\tilde{U}_1, \cdots, \tilde{U}_m\}$ on $\mathcal{G}$, where each $\tilde{U}_j$ consists of all the points matched to $\tilde{q}_j$. By Definition 6, we know that $q_j$ should be the geometric median point of $U_j$ in order to make the objective value as low as possible. Thus, we can use the well known Weiszfelds algorithm [23] to compute the geometric median point for each $\tilde{U}_j$, and update $\tilde{q}_j$ to be the corresponding geometric median point. We can iteratively perform the following two steps, (1) computing the chromatic partition and (2) generating the geometric median points, until the objective value becomes stable.

## 4  $k$-Prototype learning

In this section, we generalize the ideas for 1-prototype learning to $k$-prototype learning for some $k > 1$. As mentioned in Section 1, our idea is to build a correlation graph. We first introduce the following lemma.

**Lemma 2.** *The alignment cost in Definition 3 satisfies the triangle inequality.*

**Correlation graph.** We denote the correlation graph on the given $m$-rigid structures $\{P_1, \cdots, P_n\}$ as $\Gamma$, which contains $n$ vertices $\{v_1, \cdots, v_n\}$. Each $v_i$ represents the rigid structure $P_i$, and the edge connecting $v_i$ and $v_j$ has the weight equal to $\mathcal{A}(P_i, P_j)$. From Lemma 2, we know that $\Gamma$ is a metric graph. Thus, we have the following key theorem.

**Theorem 3.** *Any $\lambda$-approximation solution for metric $k$-median clustering on $\Gamma$ yields a $2\lambda$-approximation solution for the $k$-prototype learning problem on $\{P_1, \cdots, P_n\}$, where $\lambda \ge 1$.*

*Proof.* Let $\{Q_1, \cdots, Q_k\}$ be the $k$ rigid structures yielded in an optimal solution of the $k$-prototype learning, and $\{\mathcal{C}_1, \cdots, \mathcal{C}_k\}$ be the corresponding $k$ optimal clusters. For each $1 \le j \le k$, the cost of $\mathcal{C}_j$ is $\sum_{P_i \in \mathcal{C}_j} \mathcal{A}(P_i, Q_j)$. There exists one rigid structure $P_{i_j} \in \mathcal{C}_j$ such that

$$\mathcal{A}(P_{i_j}, Q_j) \le \frac{1}{|\mathcal{C}_j|}\sum_{P_i \in \mathcal{C}_j}\mathcal{A}(P_i, Q_j). \tag{6}$$

If we replace $Q_j$ by $P_{i_j}$, the cost of $\mathcal{C}_j$ becomes

$$\sum_{P_i \in \mathcal{C}_j}\mathcal{A}(P_i, P_{i_j}) \le \sum_{P_i \in \mathcal{C}_j}(\mathcal{A}(P_i, Q_j) + \mathcal{A}(Q_j, P_{i_j})) \le 2\sum_{P_i \in \mathcal{C}_j}\mathcal{A}(P_i, Q_j), \tag{7}$$

where the first inequality follows from the triangle inequality (by Lemma 2), and the second inequality follows from (6). Then, (7) directly implies that

$$\sum_{j=1}^{k}\sum_{P_i \in \mathcal{C}_j}\mathcal{A}(P_i, P_{i_j}) \le 2\sum_{j=1}^{k}\sum_{P_i \in \mathcal{C}_j}\mathcal{A}(P_i, Q_j), \tag{8}$$

(8) is similar to the deterministic solution in Theorem 2; the only difference is that the point-sets here need to be aligned through rigid transformation, while in Theorem 2, the point-sets are fixed.

Now, consider the correlation graph $\Gamma$. If we select $\{v_{i_1}, \cdots, v_{i_k}\}$ as the $k$ medians, the objective value of the $k$-median clustering is the same as the left-hand side of (8). Let $\{v_{i'_1}, \cdots, v_{i'_k}\}$ be the $k$ median vertices of the $\lambda$-approximation solution on $\Gamma$. Then, we have

$$\sum_{i=1}^{n} \min_{1 \leq j \leq k} \mathcal{A}(P_i, P_{i'_j}) \leq \lambda \sum_{i=1}^{n} \min_{1 \leq j \leq k} \mathcal{A}(P_i, P_{i_j}) \leq 2\lambda \sum_{j=1}^{k} \sum_{P_i \in \mathcal{C}_j} \mathcal{A}(P_i, Q_j), \qquad (9)$$

where the second inequality follows from (8). Thus the theorem is true. $\qquad\square$

Based on Theorem 3, we have the following algorithm for $k$-prototype learning.

**Algorithm: $k$-prototype learning**

1. Build the correlation graph $\Gamma$, and run the algorithm proposed in [9] to obtain a $6\frac{2}{3}$-approximation for the metric $k$-median clustering on $\Gamma$, and consequently a $13\frac{1}{3}$-approximation for $k$-prototype learning.

2. For each obtained cluster, run the 1-prototype learning algorithm presented in Section 3.

*Remark 2.* Note that there are several algorithms for metric $k$-median clustering with better approximation ratio (than $6\frac{2}{3}$), such as the ones in [19]. But they are all theoretical algorithms and have difficult to be applied in practice. We choose the linear programming rounding based algorithm by Charikar *et al.* [9] partially due to its simplicity to be implemented for practical purpose.

**The exact correlation graph is not available.** From the methods presented in Section 3.1, we know that only approximate alignments can be obtained. This means that the exact correlation graph $\Gamma$ is not available. As a consequence, the approximate correlation graph may not be metric (due to possible violation of the triangle inequality). This seems to cause the above algorithm to yield solution with no quality guarantee. Fortunately, as pointed in [9], the LP-rounding method still yields a provably good approximation solution, as long as a weaker version of the triangle inequality is satisfied (*i.e.,* for any three vertices $v_a$, $v_b$ and $v_c$ in $\Gamma$, their edge weights satisfy the inequality $w(\overline{v_a v_b}) \leq \delta(w(\overline{v_a v_c}) + w(\overline{v_b v_c}))$ for some constant $\delta > 1$, where $w(\overline{v_a v_b})$ is the weight of the edge connecting $v_a$ and $v_b$).

**Theorem 4.** *For a given set of rigid structures, if a $(1 + \epsilon)$-approximation of the alignment between any pair of rigid structures can be computed, then the algorithm for metric $k$-median clustering in [9] yields a $2(\frac{23}{3}(1 + \epsilon) - 1)(1 + \epsilon)$-approximation for the $k$-prototype learning problem.*

**What if the rigid structures have unequal sizes?** In some scenario, the rigid structures may not have the same number of points, and consequently the one-to-one match between rigid structures in Definition 2 is not available. To resolve this issue, we can use the weight normalization strategy and adopt *Earth Mover's Distance (EMD)* [8]. Generally speaking, for any rigid structure $P_i$ containing $m'$ points for some $m' \neq m$, we assign each point with a weight equal to $\frac{m}{m'}$, and compute the alignment cost based on EMD, rather than the bipartite matching cost. With this modification, both the 1- and $k$-prototype learning algorithms still work.

## 5 Exepriments

To evaluate the performance of our proposed approach, we implement our algorithms on a Linux workstation (with 2.4GHz CPU and 4GB memory). We consider two types of data, the sets of randomly generated 3D rigid structures and a real biological data set which is used to determine the organization pattern (among a population of cells) of chromosome territories inside the cell nucleus.

**Random data.** For random data, we test a number of data sets with different size. For each data set, we first randomly generate $k$ different rigid structures, $\{Q_1, \cdots, Q_k\}$. Then around each point of $Q_j$, $j = 1, \cdots, k$, we generate a set of points following Gaussian distribution, with variance $\delta$. We randomly select one point from each of the $m$ Gaussian distributions (around the $m$ points of $Q_j$) to form an $m$-rigid structure, and transform it by a random rigid transformation. Thus, we build a cluster (denoted by $\mathcal{C}_j$) of $m$-rigid structures around each $Q_j$, and $Q_j$ can be viewed as its prototype (*i.e.,* the ground truth). $\bigcup_{j=1}^{k} \mathcal{C}_j$ forms an instance of the $k$-prototype learning problem.

We run the algorithm of $k$-prototype learning in Section 4, and denote the resulting $k$ rigid structures by $\{Q'_1, \cdots, Q'_k\}$. To evaluate the performance, we compute the following two values. Firstly, we compute the bipartite matching cost, $t_1$, between $\{Q_1, \cdots, Q_k\}$ and $\{Q'_1, \cdots, Q'_k\}$, *i.e.,* build the bipartite graph between $\{Q_1, \cdots, Q_k\}$ and $\{Q'_1, \cdots, Q'_k\}$, and for each pair $Q_i$ and $Q'_j$, connect an edge with a weight equal to the alignment cost $\mathcal{A}(Q_i, Q'_j)$. Secondly, we compute the average alignment cost (denoted by $c_j$) between the rigid structures in $\mathcal{C}_j$ and $Q_j$ for $1 \leq j \leq k$, and compute the sum $t_2 = \sum_{j=1}^{k} c_j$. Finally, we use the ratio $t_1/t_2$ to show the performance. The ratio indicates how much cost (*i.e.,* $t_1$) has been reduced by our prototype learning algorithm, comparing to the cost (*i.e.,* $t_2$) of the input rigid structures. We choose $k = 1, 2, 3, 4, 5$; for each $k$, vary $m$ from 10 to 20, and the size of each $\mathcal{C}_j$ from 100 to 300. Also, for each $\mathcal{C}_j$, we vary the Gaussian variance from 10% to 30% of the average spread norm of $Q_j$, where if we assume $Q_j$ contains $m$ points $\{q_1, \cdots, q_m\}$, and $o = \frac{1}{m} \sum_{l=1}^{m} q_l$, then the average spread norm is defined as $\frac{1}{m} \sum_{l=1}^{m} ||q_l - o||$. For each $k$, we generate 10 datasets, and plot the average experimental results in Figure 2(a). The experiment suggests that our generated prototypes are much closer (at least 40% for each $k$) to the ground truth than the input rigid structures.

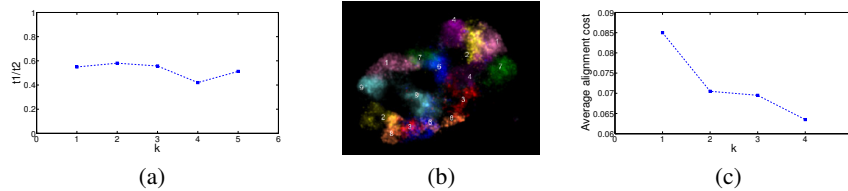

Fig. 2: (a) Experimental results for random data; (b)A 2D slice of the 3D microscopic image of 8 pairs of chromosome territories; (c ) Average alignment cost for biological data set.

**Biological data.** For real data, we use a biological data set consisting of 91 microscopic nucleus images of WI-38 lung fibroblasts cells. Each image includes 8 pairs of chromosome territories (see Fig. 2(b)). The objective of this experiment is to determine whether there exists any spatial pattern among the population of cells governing the organization of the chromosomes inside the 3D cell nucleus so as to provide new evidence to resolve a longstanding conjecture in cell biology which says that each chromosome territory has a preferable position inside the cell nucleus. For this purpose, we calculate the gravity center of each chromosome territory and use it as the representative of the chromosome. In this way, each cell is converted into a rigid structure of 16 points. Since there is no ground truth for the biological data, we directly use the average alignment cost between our generated solutions and the input rigid structures to evaluate the performance. We run our algorithms for $k = 1, 2, 3, 4$, and plot the cost in Fig. 2(c ). Our preliminary experiments indicate that there is a significant reduction on the average cost from $k = 1$ to $k = 2$, and the cost does not change too much for $k = 2, 3, 4$. We also analyze how chromosomes change their clusters when increase $k$ from 2 to 4. We denote the clusters for $k = 2$ as $\{C_1^2, C_2^2\}$, and the clusters for $k = 4$ as $\{C_1^4, C_2^4, C_3^4, C_4^4\}$. For each $1 \leq j \leq 4$, we use $\frac{|C_j^4 \cap C_1^2|}{|C_1^2|}$ and $\frac{|C_j^4 \cap C_2^2|}{|C_2^2|}$ to represent the preservation of $C_j^4$ from $C_1^2$ and $C_2^2$ respectively. The following table 1 shows the preservation (denoted by Pre) with $C_1^2$ and $C_2^2$. It shows that $C_4^4$ preserved $C_2^2$ well, meanwhile, the union of $\{C_1^4, C_2^4, C_3^4\}$ preserved $C_1^2$ well. This seems to suggest that all the cells are aggregated around two clusters.

Table 1: The preservations

| Pre | $C_1^4$ | $C_2^4$ | $C_3^4$ | $C_4^4$ |
|-----|---------|---------|---------|---------|
| $C_1^2$ | 26.53% | 18.37% | 46.94% | 8.16% |
| $C_2^2$ | 0% | 0% | 5.56% | 94.44% |

## 6 Conclusion

In this paper, we study a new prototype learning problem, called $k$-prototype learning, for 3D rigid structures, and present a practical optimization model for it. As the base case, we consider the 1-prototype learning problem, and reduce it to the chromatic clustering problem. Then we extend 1-prototype learning algorithm to $k$-prototype learning to achieve a quality guaranteed approximate solution. Finally, we implement our algorithms on both random and biological data sets. Experiments suggest that our algorithms can effectively learn prototypes from both types of data.

## Footnotes

[1] Due to space limit, we put some details and proofs in our full version paper.

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
