[Reviews · NeurIPS 2013]

Submitted by Assigned_Reviewer_4

This paper proposes an algorithm to compute k-prototypes for a set of 3D rigid structures. Two structures are called rigid, if one can align them via translation, reflection and rotation. The authors first propose a 1-prototype algorithm to find the best representative 3D structure (the one minimizing the distance to all other structure subject to optimal rigid alignement). For k-prototype clustering, the algorithm then consists of building a correlation graph and using [11] to perform k-median clustering. For each obtained cluster, the 1-prototype algorithm is applied.
The algorithm is evaluated on a synthetic example as well as on biological data.

The paper is clearly written and mathematically rigorous. Alignment and clustering of rigid structures is an important problem in eg, chromosome, protein analysis and visualization, computer vision, etc. The authors provide theoretical guarantees on the quality of the solution (although in practice, I'm not sure how much that holds since various approximation steps are taken). In most of practical applications there is typically also significant noise (where structures contain outliers) which is not really dealt with in the paper. The experiments are really sparse (with no comparison to existing techniques), so it's hard to judge the results.

Other comments:
- Algorithm 1 looks like generalize procrustes analysis. What's the relation to it?
- Given that [14] can be used to align 3D structures efficiently, performing procrustes analysis gives you the best prototype. Why not do that?
- Are there any guarantees on the search algorithm in l.188-198? Since this seems to be proposed in cases when m is large, a theoretical analysis on the complexity of the algorithm would be appropriate. At least a discussion on what is the range of m that this algorithm handles.
- At least some more visual results could be provided.
- In computer vision there are several techniques for aligning shapes. I wonder how the proposed algorithm relates/compares to those.
- Refs dealing with a similar problem (but are practical algorithms):
- Belongie et al., Shape matching and object recognition using shape contexts, TPAMI 2002
- Jian et al, Learning Shape Prior Models for Object Matching, CVPR 2009
(see also refs within)
Summary: The authors propose k-prototype clustering for 3D rigid structures. Focus is on the theoretical guarantees.

Submitted by Assigned_Reviewer_5

This paper introduces an algorithm to cluster shapes into k groups, each containing rigidly transformed versions of one shape, and computing a prototype for each group. Two new chromatic clustering algorithms are proposed to compute the prototype within a cluster with various quality guarantees.

I can see the interest of the proposed algorithm with its theoretical guarantees. However, from a practical standpoint, I am not entirely convinced of the benefits of the resulting method. In particular:
- Step 2(a) of the 1 prototype learning algorithm finds the best alignment of each shape to the current prototype. Would is really make a significant difference to simply compute the median of the n points corresponding to point q_j, instead of using the chromatic clustering algorithm? Since the alignment already guarantees a one-to-one correspondence, is it really useful to recompute it with the chromatic clustering algorithm?
- It is mentioned that ICP would not be suitable because it does not guarantee a one-to-one correspondence. This is true, but I am not sure how often this problem would occur in practice. Furthermore, since the chromatic clustering algorithm recomputes the correspondences, it may not be relevant at all.
- In practice, how important is it to run several iterations of Step 2(a) and (b)? I would expect that, if the clusters found by [11] are good, a single iteration is sufficient. If the clusters are bad, then it would be hard to find a meaningful prototype, anyway.
- While the experiments show the good behavior of the algorithm, they do not compare it against other existing methods (e.g., an ICP-based approach, or simpler versions of the algorithm as proposed above). It is thus not clear what the true benefits of the algorithm are.

It also seems to me that several references to existing shape analysis works are missing. In particular:
- The manifold of 3D shapes has been studied by Kendall, and later by Dryden & Mardia.
- A lot of work on the topic has been recently done by Srivastava and colleagues, e.g., Kurtek et al. CVPR'10.
- Other shape representations have also been studied, such as level sets (Osher & Fedkiw), or medial surfaces (Bouix et al.).

Additional comments:
- The multi-level net method seems related to a branch-and-bound approach. Would it be possible to get better guarantees with a proper branch-and-bound approach? For instance, Hartley & Kahl, IJCV'09, studied branch-and-bound to optimize rotations, although in a different context.
- In Definitions 5 and 6, why change the notation from P_i to G_i?
- Line 233, it should be m median, instead of k.
- In the synthetic experiments, were the rigid shapes transformed by random rigid transformations? With the current description, it seems that they just are noisy versions of the prototypes, which would make the task easier.
Summary: I can see the interest of having some theoretical guarantees, but I am not quite convinced about the practical benefits of the proposed method.

Submitted by Assigned_Reviewer_6

The paper addresses a new prototype learning problem with applications in in biology for determining the spatial organization pattern of chromosome territories from a population of cells.

The paper is very clearly written and I was able to follow the definitions, algorithms and proofs. Presenting the ideas for the 1-prototype learning before the k-prototype case also contributes a lot to understanding.

One point that needs clarification is:

"From Definition 4, we know that the k-prototype learning problem can be viewed as first clustering the rigid structures into k clusters and then build a prototype for each cluster so as to minimize the total alignment cost."

It is not clear to me why this holds.
Summary: A very interesting paper that presents a new problem and its solution in clear way.
Author Feedback

Author rebuttal: We thank the reviewers for their thoughtful comments. Below, we address their concerns.

Reviewer 4:

1. The relationship between Algorithm 1 and generalized procrustes analysis (GPA).

The main difference is that points from each P_i are not required to be pre-labeled in algorithm 1, while for GPA every input point should have an individual index. This is also the main difficulty for both of the alignment and the chromatic clustering steps in algorithm 1, since we do not know which point is matched to which point in advance.

2. Why not use the method in [14] (which is [13] in our camera ready version)?

As we mentioned, the algorithm in [13] is not practical, since its running time is O(m^7), and the hiding constant is also very large.

3. The time complexity of the algorithm in l.188-198.

Let c be the number of rounds that the algorithm runs, and t be the number of sub-regions it divides in each round. Then the time complexity of the algorithm is O(ctm^3), which is significantly lower than that of the algorithm in [13] (see the above question 2).

4. Comparison with other existing shape aligning algorithms.

We agree that several methods for aligning shapes exist, but most of them are not suitable for our problem. For example, the idea of the matching algorithm via shape contexts by Belongie et al. is to build a histogram for each point, but it requires that the distribution of the points be dense enough, otherwise the histogram would be meaningless; however, in our problem the distribution of the points from each rigid structure could be quite sparse. The algorithm by Jiang et al. is for non-rigid transformation or affine transformation, rather than rigid transformation on which our paper mainly focuses. Their method uses QR decomposition to find the optimal affine transformation matrix, which is challenging to tailor it to suit the rigid transformation in our problem.

Reviewer 5:

1. Is chromatic clustering necessary?

Yes, it is. As pointed out by the reviewer, simply computing the median of the n points corresponding to each q_j indeed yields a feasible solution. But it is not optimal with respect to the cost function. On the contrary, chromatic clustering seeks to obtain the minimum cost for the new configuration in each round.

2. How often the one-to-one correspondence occurs in practice. Furthermore, since the chromatic clustering algorithm recomputes the correspondences, it may not be relevant at all.

-The one-to-one correspondence is necessary for some applications. For example, the biology problem studied in this paper requires that every chromosome territory be uniquely matched to another chromosome territory in the other cell. Many other pattern matching problems also have such a requirement, especially for those in which no two feature points can be matched to the same one (i.e., mutual exclusiveness exists among feature points).

-For the latter question, the final result is relevant to the one-to-one correspondence requirement in the alignment step, even though the following chromatic clustering step can also ensure one-to-one correspondence. This is because our purpose is not only obtaining the one-to-one point matching between the prototype and each rigid structure, but also minimizing the objective value. The performance of chromatic clustering depends on the configuration of the rigid structures in the previous alignment step, and incorrect alignments for some rigid structures could cause poor performance (i.e., high objective value) for the chromatic clustering.

3. How important is it to run several iterations of step 2(a) and 2(b)?

The clustering from the algorithm in [11] (which is [9] in our camera ready version) can only guarantee a constant approximation (by theorem 3), but may not be good enough. To further reduce the objective value, we need to run 2(a) and 2(b) iteratively.

4. Why not use branch-and-bound (BB) approach.

BB approach needs to grow a searching tree in the parameter space, and for each node it requires estimating the upper and lower bounds of the objective value in the corresponding sub-region. But for our alignment problem, it is challenging to obtain such accurate estimations. Moreover, BB approach grows the searching tree in all the sub-regions not satisfying the pruning conditions, which could make the running time grow exponentially.

5. Why change P_i to G_i in Definition 5 and 6?

Actually, when rigid structure P_i is fixed in the space, it is equivalent to G_i. We use different notations to distinguish rigid structure from static point-set.

6. Were the rigid structures transformed by random rigid transformation in the synthetic experiments?

Yes, they were. We should mention it more clearly in the paper, and thank the reviewer for pointing it out.

Reviewer 6:

1. Clarify ``From Definition 4, ……………minimize the total alignment cost."

From formula (1) in Definition 4, we know that if each P_i is assigned to the one in {Q_1, …, Q_k} with the minimum alignment cost with P_i, then all rigid structures are automatically divided into k clusters, and each Q_j is the prototype of its corresponding cluster. But in reality, since {Q_1, …, Q_k} are unknown in advance, we need to first cluster {P_1, …, P_n} into k clusters based on their pairwise similarities, and then independently build the prototype for each cluster. We are able to show that although clustering the rigid structures (without knowing {Q_1, …, Q_k}) may not yield the optimal solution, it produces an approximation solution with quality guarantee (see Theorem 3).